# Modeling and Evaluation of Penetration Process Based on 3D Mechanical Simulation

**DOI:** 10.3390/s24216988

**Published:** 2024-10-30

**Authors:** Xiaohan Chen, Huiying Gong, Bin Yang, Zengshuo Wang, Yaowei Liu, Lu Zhou, Xin Zhao, Mingzhu Sun

**Affiliations:** 1National Key Laboratory of Intelligent Tracking and Forecasting for Infectious Diseases, Engineering Research Center of Trusted Behavior Intelligence, Ministry of Education, Tianjin Key Laboratory of Intelligent Robotics, Institute of Robotics and Automatic Information System, Nankai University, Tianjin 300350, China; stuchenxiaohan@163.com (X.C.); 2120190360@mail.nankai.edu.cn (H.G.); 2120210384@mail.nankai.edu.cn (B.Y.); zengshuowang@mail.nankai.edu.cn (Z.W.); liuyaowei@mail.nankai.edu.cn (Y.L.); zhoulu@nankai.edu.cn (L.Z.); zhaoxin@nankai.edu.cn (X.Z.); 2Institute of Intelligence Technology and Robotic Systems, Shenzhen Research Institute of Nankai University, Shenzhen 518083, China

**Keywords:** micromanipulation, intracellular stress, cell penetration, finite element simulation, force and deformation

## Abstract

In biological micromanipulation, cell penetration is a typical procedure that precedes cell injection or oocyte enucleation. During this procedure, cells usually undergo significant deformation, which leads to cell damage. In this paper, we focus on modeling and evaluating the cell penetration process to reduce cell deformation and stress, thereby reducing cell damage. Initially, a finite element model (FEM) is established to simulate the cell penetration process. The effectiveness of the model is then verified through visual detection and comparison of cell deformation with experimental data. Next, various mechanical responses are analyzed, considering the influence of parameters, such as the radius and shape of the injection micropipettes, material properties, and size of the cells. Finally, the relationship between the intracellular stress and the cell penetration depth of biological cells is obtained. The evaluation results will be applied to develop optimized operation plans, enhancing the efficiency and safety of the cell penetration process.

## 1. Introduction

Somatic cell nuclear transplantation (SCNT), commonly known as cloning, has extensive applications in gene therapy, organ transplantation, animal breeding, and other fields. A critical step in SCNT is cell penetration, which exerts forces that significantly impact the developmental potential of the operated cells. Cell penetration plays an important role in the treatment of tumor treatment [1,2], ocular neovascular diseases [3], and other conditions. Reducing cell damage during this manipulation is a crucial challenge in micromanipulation research, which motivates us to evaluate both the intracellular stress involved and the resultant damage during cell penetration.

Several research groups have proposed various mechanical measurement methods. Abadie et al. developed a force sensor made up of a magnetic spring. The sensor was applied to oocytes to calculate the force by measuring the compression length of the magnetic spring [4]. Sun et al. attached a set of micropipettes to a microelectromechanical force sensor to measure force–deformation curves during the penetration [5]. Although these methods accurately reflect the forces acting on cells, they focus more on overall force analysis and pay less attention to the distribution of intracellular stress. However, obtaining intracellular stress and the corresponding models is more meaningful for cell penetration due to the complexity of cellular structures. Considering the difficulty in theoretical modeling and the challenge in obtaining data for experimental modeling, it is more reasonable to establish a force model within the cell using simulation.

Mechanical models of cell manipulation can generally be divided into three categories: The first category includes point–load models [5] and static equilibrium equation models [6], which relate deformations to applied forces through mechanical equations. The second category comprises equivalent cytoskeleton models [7] and dissipative particle dynamics models [8], which establish relationships between cell loading and deformation. The third category is finite element modeling (FEM), which treats the cell as an elastic body and investigates the intracellular strain distribution during penetration. Although the first two categories of models simulate successful cell manipulation, their inherent constraints prevent them from addressing issues such as large deformations or ruptures. Furthermore, most models focus on the cell structure itself, without adequately considering the influence of external factors on manipulation. The finite element model, on the other hand, can realize the above functions and operates at a fast speed. Therefore, we employ the third modeling method, finite element simulation, to establish the model.

This paper presents a FEM that effectively simulates cell penetration and explores the mechanical influence on cells during this process. First, a mechanical model of cell penetration is established in the ANSYS environment. Subsequently, the model is verified by visually comparing cell deformation parameters obtained from simulations with those from biological experiments. Next, to generalize the model, we conduct a systematic analysis of the factors that affect cell penetration experiments, including micropipette shape and radius, cell size, material properties, and the components of the applied load. Simulations are performed to investigate how these simulation parameters influence the stress acting on cells. We reveal the relationship between intracellular stress and cell penetration depth, providing an optimization strategy aimed at enhancing cell preservation and developmental potential during surgical procedures.

## 2. Materials and Methods

### 2.1. Mechanical Modeling of Cell Penetration Process

#### 2.1.1. Cellular Structure Setting

As shown in Figure 1a, the cell model comprised two main components: the cytoplasm and the zona pellucida (ZP). We determined the ZP’s thickness and the oocyte’s radius by image processing of actual oocytes. Circle fitting was employed to determine the radius of both oocyte and cytoplasm. The ZP’s thickness was then calculated by subtracting the radius of the cytoplasm from that of the oocyte. We revealed an average oocyte radius of 80 ± 5 µm and a ZP thickness of 20 ± 5 µm. ANSYS Student software (ANSYS Student 2024 R2) was applied to model the oocyte, as shown in Figure 1b.

#### 2.1.2. Cellular Material Setting

We adopted the shell model [9] and the half–space model [10], presupposing the cell to be a linear, elastic, and isotropic material. Based on our preliminary measurements, we set Young’s modulus of the cytoplasm as 784 ± 50.3 Pa and that of the ZP as 20,000 ± 5000 Pa [11]. Considering the near–incompressibility of cellular structures (with a Poisson’s ratio approximating 0.5), the Poisson’s ratios of both cytoplasm and thylakoid are set to 0.499. Moreover, given that Young’s modulus of micropipettes is significantly higher than that of the cell, we modeled the micropipette as a rigid body.

#### 2.1.3. Penetration Setting

Displacements of the micropipette were applied to the cells along the negative direction of the X-axis. Displacements were incremented in steps of 5 µm, starting from the center of the ZP surface. The intracellular stress was calculated based on the point–load model.

#### 2.1.4. Constraint Setting

(1)Grid Setting

Tetrahedral meshing was used for both cytoplasm and ZP, with a grid cell size of 7 µm. Since microneedles are not the object of study, no meshing was performed.

(2)Contact Settings

We assumed that the contact between the holding micropipette and the ZP, as well as between the ZP and the cytoplasm, was bonded. The contact between the injection micropipette and the ZP was set as friction contact, with a friction coefficient of 0.3. To simplify the model, we assumed that the role of the holding micropipette is limited to stabilizing the cell’s position, without considering its actual impact on the cellular deformation.

(3)Load Setting

The left surface of the holding micropipette was fixed horizontally. In the initial analysis step, a 1 μm horizontal displacement was applied to the left on the injection micropipette to ensure stable contact between the micropipette and the cell. In the second step, a 5 µm displacement was applied in the same direction, followed by sequential increments of 5 µm for subsequent load steps. Intracellular stress was measured over a range of penetration depths from 1 to 130 µm.

### 2.2. Vision–Based Cell Deformation Parameter Detection

As the injection micropipette penetrated the cell, the cell underwent overall deformation due to compression, i.e., being compressed in the X direction and elongated in the Y direction. In this study, we proposed a vision–based method for detecting cell deformation parameters. We then compared the deformation parameters obtained from model simulations with experimental results to verify the accuracy of the model.

#### 2.2.1. Cell Region of Interest Extraction

We first extracted the region where the cell was located in the image sequence of cell penetration. Considering that the cell was circular in the image before contact with the injection micropipette, we obtained the cell region in the first frame of the sequence image by using Hough circle detection. Subsequently, during the cell penetration process, we cropped an appropriate sub–image as the region of interest for the cell based on the positioning results of the cytoplasm. Taking into account the deformation of the cell, we appropriately increased the length of the region in both X and Y directions.

#### 2.2.2. Cell Contour Detection

The original cell images in the experiment and simulation are shown in Figure 2a and Figure 2d, respectively. First, we obtained the cell edge image using Canny edge detection [12]. The high and low threshold values are set as 100 and 200, respectively. Then, we performed a morphological close operation [13] on the edge images, with a window size of 20 × 20 pixels to connect the contours (Figure 2b,e). Finally, we eliminated the small outlines from the images by applying an area threshold and get the cell contours, which is the red curve showed in Figure 2c,f.

#### 2.2.3. Geometric Parameters Calculation

First, we obtained the minimal bounding box of the cell contour. The green box is the bounding box, as showed in Figure 2c,f. Then, we established the unit conversion ratio by comparing the actual cell radius with the detected circle radius. Next, we calculated the width reduction value of the bounding box and invaginate value of the cell contour as the cell deformation parameters. Finally, we sorted the data by penetration depths and saved all the results.

### 2.3. Analysis of Factors Influencing the Penetration Process

Oocytes differed in size and elastic properties, as do the size and shape of the micropipettes used. These factors significantly influenced the results of the experiment. We adapted the model to a wider range of situations to enhance its versatility. To further investigate the accuracy of the model and the effects of other external conditions on intracellular stress, we conducted five sets of comparative simulations. These simulations varied the size and shape of the injection micropipettes, the size and material properties of the oocytes, and the components of the applied load during the simulations. The factors influencing the simulations are illustrated schematically in Figure 3.

#### 2.3.1. Effect of Micropipette Shape

Multiple kinds of micropipettes were used in the simulation models and experiments, featuring different tip shapes suitable for various applications. To investigate the influence of micropipette shape on experimental results, we designed four types of micropipettes for penetration simulations. The micropipette models and their corresponding cross–sections are illustrated in Figure 4. In actual cell manipulation, the micropipette resembled a thin tube with a tilted tip and a hollow interior. We constructed a micropipette model, as shown in Figure 4a, to closely match the experimental micropipette. Additionally, we created a solid column micropipette with a tilted tip to enhance the experimental outcomes, as shown in Figure 4b. In Sun’s finite element model, the micropipette tip was designed as a central protrusion of a symmetric solid column along the X-axis (Figure 4c) [14]. Furthermore, the micropipette was simplified to a flat tip of the hollow thin tube in the point–load model (Figure 4d).

Models (a) and (b) (tilted tip micropipettes) were commonly utilized in SCNT for oocyte enucleation, as shown in Figure 5a. In penetration simulation, when the radius difference between the injection micropipette and the cell was small, using a micropipette with a tilted tip resulted in uneven force distribution on the cell, as shown in Figure 5b. This result was inconsistent with the modeling assumption of applying uniform hydrostatic pressure on the cell. However, the tilted tip micropipette was suitable for modeling the relationship between the position of the tip inside the oocyte, and the positions of the nuclear and the polar body [15]. Hence, it was necessary to model the microneedle with this shape for actual oocyte enucleation.

Model (c) (sharp micropipette) more easily penetrated the cell membrane, as shown in Figure 6a. Figure 6b illustrates the cell cross–section during the penetration process. The radius of the cell indentation increased sequentially from right to left, forming an overall symmetrical cone–shaped distribution, with uniform force applied in the horizontal direction. To penetrate the cell and maintain uniform cell force in FEM simulations, the micropipette model was designed as Model (c) with a symmetrical tip shape.

Model (d) (flat–tipped micropipette) was applied to analyze cases where the penetration depth was large and the effect of the micropipette tip can be ignored. This simplification facilitated the fitting of the relationship between cell deformation and the forces exerted during the penetration process. In the point–load model, Sun omitted the impact of the micropipette tip in the experiment and used Model (d) to find that the stress value was related to the radius of the penetrating micropipette [5]. Additionally, in both the static equilibrium equation model and the dissipative particle model, the micropipette was also approximated as a cylinder with a uniform radius. In FEM simulation, even when large loads were applied, intracellular stress–penetration results were obtained using the micropipette of Model (d), allowing for a more effective and accurate analysis of the cell penetration process.

#### 2.3.2. Effect of Cell Size

Cells of different types, or even the same type of cell measured in different ways, varied in size. For instance, the largest cell in the human body is the mature egg, with a diameter of up to 200 µm. In contrast, the smallest cell, such as the platelet, has a diameter of only 2 µm. Chen utilized the Laguerre and mosaic model to investigate the effect of cell size changes on the compressive and shear strength of closed–cell foams. His findings suggested that both compressive and shear strengths increased with the growth of cell size parameters and cell size dispersion parameters [16]. It was inferred that larger cells result in higher compressive and shear strengths compared to smaller cells when penetrated to the same depth. Therefore, the impact of cell size was examined in simulations.

#### 2.3.3. Effect of Microinjection Radius

Stress was related to the size of the injection micropipette, according to the intracellular stress calculation formulas from various models, including the point–load model, static equilibrium equation model, and dissipative particle model. For instance, in the point–load model, the applied force F was expressed in Equation (1):(1)$$\begin{array}{c} \;\;\;\;\;\;\;\;\;\;\;\;\;\;\;\;\;\;\;\;\;\;\;\;\;\;F=\frac{2\pi Ehw_{d}^{3}}{a^{2}(1-v)}\left[\frac{3-4\zeta^{2}+2ln\zeta^{2}+\zeta^{4}}{\left(1-\zeta^{2}\right){\left(1-\zeta^{2}+ln\zeta^{2}\right)}^{3}}\right] \end{array}$$
where $w_{d}$ is the depth of the cellular depression, a is the radius of the depression, c is the radius of the microinjection, ζ=c/a. E is the membrane elastic modulus, v is the Poisson ratio, and h is membrane thickness. A larger ζ value within a certain range (commonly used micropipette sizes) resulted in a nonlinear increase in intracellular stress, assuming other variables remained constant.

As shown in Figure 7a, when the radius of the micropipette was much smaller than that of the cell, deformation was localized around the tip of the micropipette. As the micropipette radius increases, the deformation became more extensive, as shown in Figure 7b. It suggests that intracellular stress increased with the radius of the micropipette when penetrating to the same depth. However, a thinner micropipette did not necessarily result in lower intracellular stress or less cellular damage. If the micropipette is too thin, it might lack sufficient rigidity, leading to bending and deviation from the central axis, which hindered effective cell penetration. When the micropipette width reached a certain range, as illustrated in Figure 7c,d, the area over which force was applied increases for the same penetration depth, and the overall deformation change gradually decreased, inferring that the cell deformation stabilized when the injection micropipette size was large enough.

#### 2.3.4. Effects of Cellular Viscoelastic Plasmas

Numerous studies suggested that cells exhibit viscoelastic properties [17,18]. The linear elastic material model limited the deformation properties of cells as penetration depth increased. However, actual cells tended to experience a minor rebound after penetration, with the degree of rebound varying according to Young’s modulus of the cells. Chen et al. analyzed this creep behavior of cells using a force–clamping technique in the atomic force microscope (AFM) [19]. In this study, we investigated the effect of cellular materials, including linear elastic and viscoelastic properties.

#### 2.3.5. Effect of Load Components

In the previous section, the failure of nonlinear convergence occurred when the load was applied to the micropipette using the viscoelastic material settings. To investigate the effect of load components, we constructed two different forms of the ZP model with the corresponding contact method, as shown in Figure 8a. Here, the ZP was divided into three parts: two cylindrical sections and a remaining tapered section. The radius of the cylindrical sections was the same as that of the micropipette. These three parts were bound together. In Figure 8b, the load component was the micropipette surface in contact with the cell, while in Figure 8c, the load component was the surface where the ZP overlapped with the micropipette. In both cases, the load direction for the structures aligned with the negative direction of the X-axis.

## 3. Results

### 3.1. Preparation of Experimental Materials

The preparation process for oocytes used in the penetration experiment was as follows:(1)Ovaries were collected from slaughterhouses and transported to the laboratory in a thermos containing physiological saline at the temperature of 35 °C to 37 °C. The transportation was completed within two hours. Upon arrival, the ovaries were immediately washed twice with sterile physiological saline at 37 °C, which contains 100 IU/L penicillin and 50 mg/L streptomycin.(2)Oocytes were extracted from follicles with a diameter of 2 to 6 mm. The extracted oocytes were rinsed three times with TL–Hepes–PVA solution and then placed in an incubator at 39 °C with a carbon dioxide concentration of 5% for in vitro maturation (IVM) for 42 h.(3)After maturation, the oocytes were removed from the incubator and treated with 0.1% hyaluronidase. The oocytes were then washed three times with M199 solution. The processed oocytes were prepared for use in the penetration experiment. Some cells were freshly obtained from breeding institutions, stored in room temperature water, and used directly for experiments within 4 to 6 h.

### 3.2. Cell Penetration Experiments

Cell penetration experiments were performed using a micromanipulation system, as shown in Figure 9. The system was based on an inverted microscope (Olympus, IX–53, Tokyo, Japan) equipped with a motorized X–Y stage. A 35 mm petri dish filled with cell samples was placed on the X–Y stage. The system included a pair of motorized X–Y–Z micromanipulators. The left micromanipulator (Sutter Instrument, MP285A, Novato, CA, USA) was used for precise control of the holding micropipette, while the right micromanipulator (Scientifica, PatchStar, Uckfield, UK) was used to operate the injection micropipette. A pneumatic injector provided negative pressure for holding and positive pressure for injection, with the opening adjustment and injection pressure controlled by a stepper motor. A CCD camera (Basler, acA640–120 gm, Ahrensburg, Germany) was employed to capture images of the cells under the microscope.

The specific steps for the penetration experiment are as follows: First, place the petri dish on the X–Y stage of the micromanipulation system. Attach the holding micropipette and the injection micropipette to the left and right micromanipulators, respectively. Then, adjust the height of the manipulators to ensure that both micropipettes are in the same focal plane. Next, adjust the microscope’s field of view so that the target oocyte is centered. Apply negative pressure to the holding micropipette to stabilize the oocyte. Finally, allow the injection micropipette to advance along the central axis of the oocyte at the preset speed until it breaks the cell membrane.

The penetration depths ranged from 1 to 130 µm, with increments of 5 µm. For each experimental penetration depth, we captured the image sequence of the cell penetration at various penetration depths and performed offline analysis of the deformation and stress during the penetration process.

### 3.3. Simulation Model Verification Based on Cell Deformation Detection

Before conducting mechanical simulations, we first validated the effectiveness of the simulation model. We performed visual detection for both FEM–based penetration simulation and the actual cell penetration experiment, extracting the width reduction value and the invaginate value of the cell during the penetration process as deformation parameters, as shown in Figure 10. The simulation model was verified by comparing the relationships between the penetration depth and the deformation parameters.

In FEM simulations, the simulation parameters were set according to the actual cell penetration experiment as follows: Young’s modulus of the cytoplasm was 784 Pa with a radius of 55 μm. Young’s modulus of the ZP was 20,000 Pa, with an outer radius of 75 μm and a thickness is 20 μm. The range of penetration depth was set from 1 to 130 µm, and the injection micropipette had a radius of 10 µm. In cell penetration experiments, due to the good consistency between the penetration depth and actual cell deformation data, we randomly selected three oocytes for model verification.

Figure 11 shows the comparison between the simulation and experimental results. The horizontal axis represents the penetration depth into the cell, while the vertical axis represents cell width reduction along the X-axis (Figure 11a), and the cell invaginate value (Figure 11b), respectively. The width reduction values and the invaginate values calculated in the model simulation were very close to the results of the cell penetration experiment. The changes in cell invagination were found to be opposite to those in width reduction. Moreover, we performed a correlation fit between the cell deformation data from the FEM simulation and that from the penetration experiment using Pearson fitting method. The fitting results are shown in Table 1. For different experimental results, the correlation between the penetration depth–deformation curve of the simulation and the experiments exceeded 0.99.

In summary, the simulation model accurately reflected the relationship between penetration depth and cell deformation during the penetration process, which indicates that the model effectively simulated the experimental process and yielded precise simulation data.

### 3.4. Mechanical Simulation of Cell Penetration Process Under the Influence of Multiple Factors

#### 3.4.1. Simulation Model Testing

In simulations, the Young’s modulus of the cytoplasm was set as 784 Pa with a radius of 55 µm. The Young’s modulus of the ZP was set as 20,000 Pa, with an outer radius of 75 µm and a thickness of 20 µm. The holding micropipette had an outer radius of 40 µm and an inner radius of 20 µm. The injection micropipette had a radius of 10 µm. The penetration depth of the injection micropipette ranges from 1 to 130 µm. Figure 12 illustrates the simulation procedure for cell penetration, and Figure 13 presents the stress–displacement curve derived from the simulation data. The X-axis represents the cell penetration depth, while the Y–axis indicates the intracellular stress. The intracellular stress increased nonlinearly with the change in penetration depth. When the depth reached 75 µm, both the ZP and the cell membrane were pierced, getting the measured maximum force, which was 6.3 µN.

#### 3.4.2. Mechanical Simulation Results of Cell Penetration under the Influence of Multiple Factors

(1)Influence of injection micropipette shape

To investigate the effect of the shape of the injection micropipette, four different micropipette designs were constructed based on the settings established in Section 3.4.1. Figure 14 illustrates the Stress–penetration depth curves obtained from the FEM for four different micropipettes.

The overall trends of intracellular stress were similar for the micropipettes with different tip shapes. As shown in Table 2, when cells were penetrated to the same depth, the intracellular stress showed little change across all micropipette shapes. An increase in penetration depth corresponded to an increase in intracellular stress, which was consistent with previous experimental results.

In simulations, micropipettes of different shapes could be utilized in various simulation scenarios, resulting in accurate outcomes. However, in actual cell manipulations, the tilted tip solid micropipette (Model (b)) and sharp micropipette (Model (c)) were not suitable for enucleation. The flat–tipped micropipette (Model (d)) induced greater intracellular stress and could not use for cell penetration compared to the tilted tip hollow micropipette (Model (a)) at the same penetration depth. Overall, the tilted tip hollow micropipette (Model (a)) successfully performed most cell penetration manipulations with minimal stress. We mainly used this type of micropipette in the experiments considering time and cost.

(2)Influence of cell size

To investigate the effect of cell size, the inner and outer radii of the cell were varied based on the model setup in Section 3.4.1. The outer radius of the cell, i.e., the radius of ZP, denoted as *R*, was set to 70 µm, 75 µm, and 80 µm, respectively. The inner radius of the cell, i.e., the radius of the cytoplasmic, was set to 50 µm, 55 µm, and 60 µm, respectively.

As shown in Figure 15, the overall trends of intracellular stress were similar across different cell sizes. However, as the cell size increased, the intracellular stress decreased for the same penetration depth, as shown in Table 3. Since cells were interconnected by bonds in the simulation, an increase in stress led to the breaking of more bonds, leading to greater cellular damage [8]. Therefore, we can select larger cells in actual manipulations to improve cell survival rates after cell manipulations.

(3)Influence of micropipette radius

To investigate the effect of micropipette radius, different sizes of micropipettes were set based on the setup in Section 3.4.1. As shown in Figure 16a, the radius of the injection micropipette was set to 5 µm, 7.5 µm, 10 µm, and 12.5 µm, respectively. We employed the flat–tipped micropipette (Model (d)) as the micropipette model and set the cytoplasmic radius to 55 µm, which was the median value of the radius of the cytoplasm radius range.

As shown in Figure 16b, the intracellular stress for micropipettes with different radii was quite similar. However, the intracellular stress increased with the increasing radius of the micropipette, as shown in Table 4. A larger micropipette radius resulted in the breaking of more bonds and more severe cell damage. Therefore, it was suggested to use a micropipette with a smaller radius to minimize mechanical damage. Nevertheless, as the micropipette radius decreased, the micropipette could bend in the middle, resulting in penetration failures. We will discuss how to determine the appropriate micropipette radius in the following section.

(4)Influence of cellular viscoelastic material

To investigate the effect of cellular materials, two types of materials with linear elastic and viscoelastic properties were set based on the setup in Section 3.4.1. For the linear elastic cytoplasm, Young’s modulus was set to 784 Pa, while that for the ZP is 20,000 Pa. For the viscoelastic cytoplasm, Young’s modulus was set to 3160 Pa, and it was 17,630 Pa for the ZP [15]. Based on the previous section, we selected micropipettes with radii of 10 µm and 7.5 µm, both being flat–tipped micropipettes, for the penetration simulations.

As shown in Figure 17 and Table 5, cells with viscoelastic materials exhibited greater stress than those with linear elastic materials when penetrated by micropipettes of the same size to the same depth. Under the same material conditions, the stress variation was more pronounced for viscoelastic materials across different micropipette sizes at the same penetration depth. Regarding cell deformation, as illustrated in Figure 18, the simulation results for viscoelastic materials were closer to the experimental data. Thus, it was more accurate to model cellular material as viscoelastic in simulations. However, due to the technical challenges and the complexity of certain mathematical problems, linear elastic materials may be chosen for more complex cellular operations.

(5)Influence of load components

Different loads were set up for simulations based on the viscoelastic settings in the previous section. The load components were located on the injection micropipette and the cell region overlapping with the injection micropipette, respectively, along the negative direction of the X-axis.

As shown in Figure 19 and Table 6, when loads were applied to either the cell or the injection micropipette, there was little difference in both intracellular stress and strain energy for the same–sized injection micropipettes at the same penetration depth. Regarding cell deformation, as illustrated in Figure 20, the curves representing cell width reduction and invagination at the same penetration depth but with different load components, also coincided. Therefore, the load component had little impact on the simulation results. It is reasonable to apply the load to the cell in the viscoelastic settings.

#### 3.4.3. Intracellular Stress Modeling During Penetration

Based on the FEM simulation results, we further developed an intracellular stress model that related to micropipette radius and cell radius. For a specific type of cell with a radius of 75 µm, we conducted penetration simulations using six micropipettes of varying radii (r = 5, 7.5, 9, 10, 12.5, and 15 µm) according to the model setup in Section 3.4.1. Specifically, the penetration depths were set along the X-axis near the tip of the micropipette, ranging from 1 µm to 130 µm in increments of 5 µm. Figure 21 shows the force exerted on the cell by various loads for different micropipette radii. Each line in the figure represents a control simulation performed with a micropipette radius.

We performed a nonlinear regression analysis on the simulation results in Figure 21. The following expected function was chosen for its continuity and differentiability:(2)$$\begin{array}{c} \;\;\;\;\;\;\;\;\;\;\;\;\;\;\;\;\;\;\;\;\;\;\;\;\;\;F=\alpha d^{3} \end{array}$$
where F is the intracellular stress during the penetration process, α  is the fitting parameter, d is the penetration depth. The fitting parameters  α  for micropipettes of different radii are shown in Table 2.

As shown in Table 7, the injection micropipette radius r had a significant effect on the fitting parameter α. The relationship between the r and the α was characterized by a power–law fit. The fitting results are shown in Equation (3) and Figure 22:(3)$$\begin{array}{c} \;\;\;\;\;\;\;\;\;\;\;\;\;\;\;\;\;\;\;\;\;\;\;\;\;\;\alpha =5.613\times 10^{-6}r^{0.4997} \end{array}$$

When the cell radius was 75 µm, the intracellular stress F at a given penetration depth d for different injection micropipette radii r could be expressed as:(4)$$\begin{array}{c} \;\;\;\;\;\;\;\;\;\;\;\;\;\;\;\;\;\;\;\;\;\;\;\;\;\;F=5.613\times 10^{-6}r^{0.4997}d^{3} \end{array}$$

Both the cell size *R* and the micropipette radius *r* influenced the intracellular stress during cell penetration. We further considered the impact of the cell radius and conducted simulations of three groups of cells with different sizes (*R* = 70, 75, and 80 µm, with the ZP thickness of 20 µm in all cases). Equation (5) shows the general form of the intracellular stress model:(5)$$\begin{array}{c} \;\;\;\;\;\;\;\;\;\;\;\;\;\;\;\;\;\;\;\;\;\;\;\;\;\;F=\alpha 1r^{\beta }d^{3} \end{array}$$
where α1 and β are the fitting parameters. Table 8 shows the fitting results of parameters α1 and β. The fit goodness was approximately 0.99 for all cases, indicating the adaptability of the model.

In this study, we analyzed the factors influencing the cell penetration process. Cell penetration further affected the developmental potential of the cells. In our previous studies, we found that as the penetration speed increased, the maximum intracellular strain of cells decreased. Additionally, the cleavage rates and the expression levels of totipotency and antiapoptotic genes of embryos also increased significantly [11]. This study implies that the development potential of cells may be inferred from the deformation change of cells during penetration. In this paper, we established the relationship between intracellular stress and cell penetration depth through FEM simulation, allowing us to analyze the force exerted on cells by observing their deformation in biological experiments. In the future, we will explore the developmental potential by analyzing cell stress based on the previous experimental results.

## 4. Conclusions

In this paper, we presented a cell penetration model based on the three–dimensional mechanical simulation and evaluated the intracellular stress during penetration using this model. The finite element method was employed to simulate the cell penetration process under various parameters, including the shape and radius of the micropipette, the size and material of the cell, and the components of the applied load. Additionally, a visual–based method for deformation parameter detection was utilized to validate the accuracy of the model.

The simulation results indicated that micropipettes of different shapes could be applied to various simulated scenarios. When using the tilted tip micropipette (Model (a) in Section 2.3), the cell experienced uniform force during penetration with minimal damage. Therefore, it is recommended to use Model (a) for most cell operations, thereby enhancing the accuracy of the simulation model while considering operational effectiveness and cost. We also found that modeling the cell material as viscoelastic was more realistic. However, it posed significant technical challenges and complex mathematical processing. In practical modeling, linear elastic materials are often chosen as a feasible alternative.

Moreover, based on the mechanical simulation, we developed an intracellular stress model related to the radius of the micropipette and cell size. A reasonable radius range of the micropipette can be determined by calculating the intracellular stress at various penetration depths using Equation (5). Notably, the accuracy of the model exceeded 99% for oocytes.

## Figures and Tables

**Figure 1 sensors-24-06988-f001:**
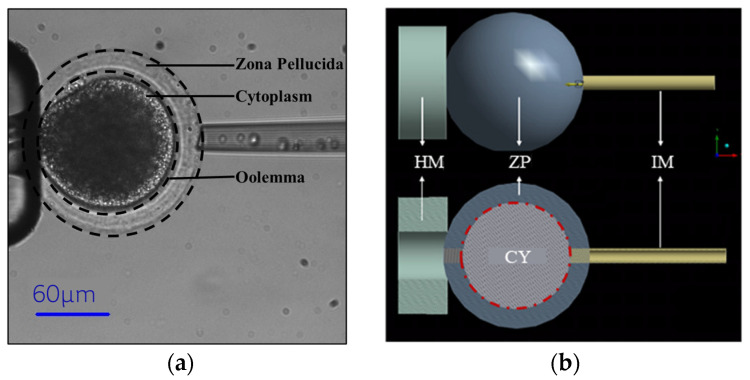
(**a**) Oocyte structure. (**b**) 3D FEM with X–Y section (The HM is the holding micropipette that holds cells and stabilizes them. The IM is the injection micropipette that is used for penetration).

**Figure 2 sensors-24-06988-f002:**
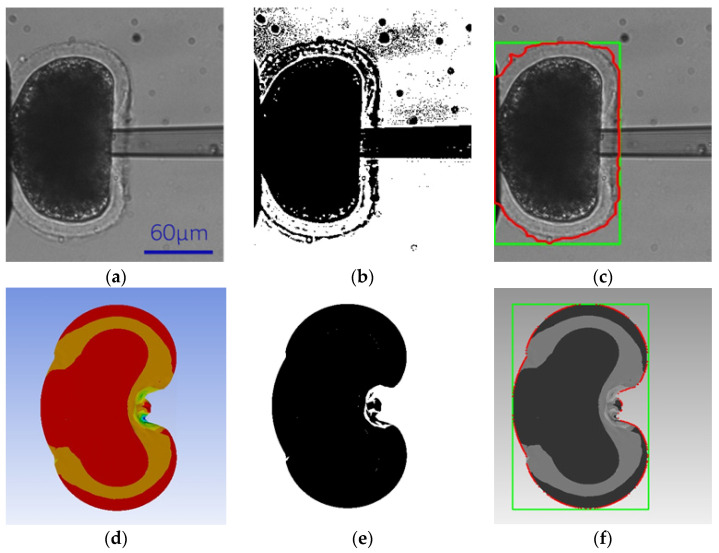
(**a**–**c**) Experimental image processing process. (**d**–**f**) FEM image processing process.

**Figure 3 sensors-24-06988-f003:**
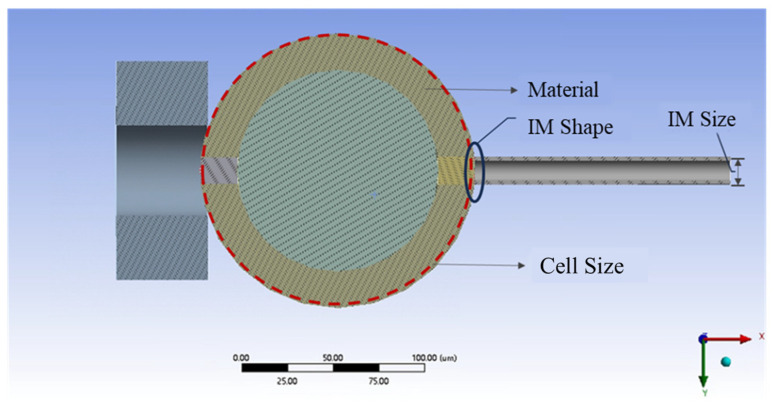
Schematic diagram of factors influencing the simulations.

**Figure 4 sensors-24-06988-f004:**
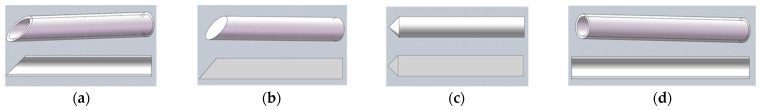
(**a**) Tilted tip hollow micropipette (**b**) Tilted tip solid micropipette (**c**) Sharp micropipette (**d**) Flat–tipped micropipette.

**Figure 5 sensors-24-06988-f005:**
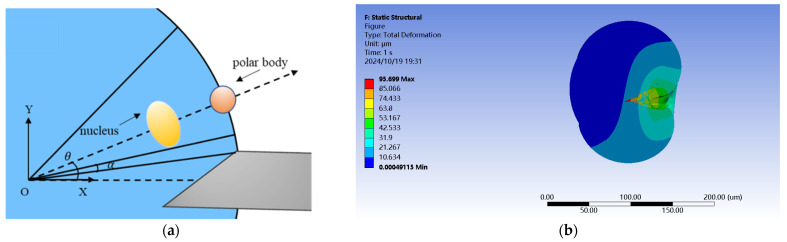
(**a**) Schematic diagram of enucleation using micropipette. (**b**) Cross–section of simulation results with micropipettes of Models (a) and (b).

**Figure 6 sensors-24-06988-f006:**
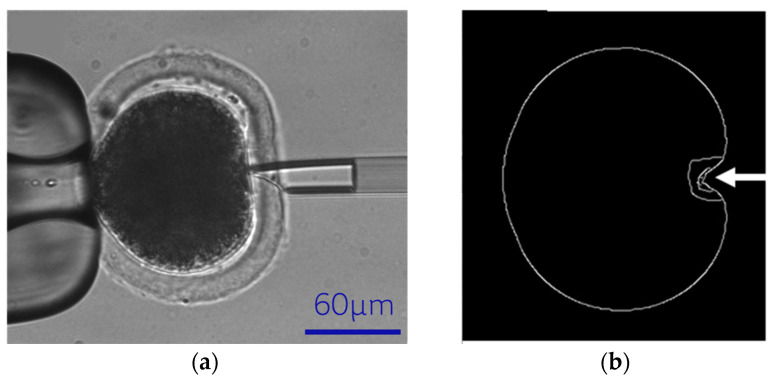
(**a**) Experimental image of cell penetration. (**b**) Oocyte contour during penetration by using the micropipette of Model (c).

**Figure 7 sensors-24-06988-f007:**
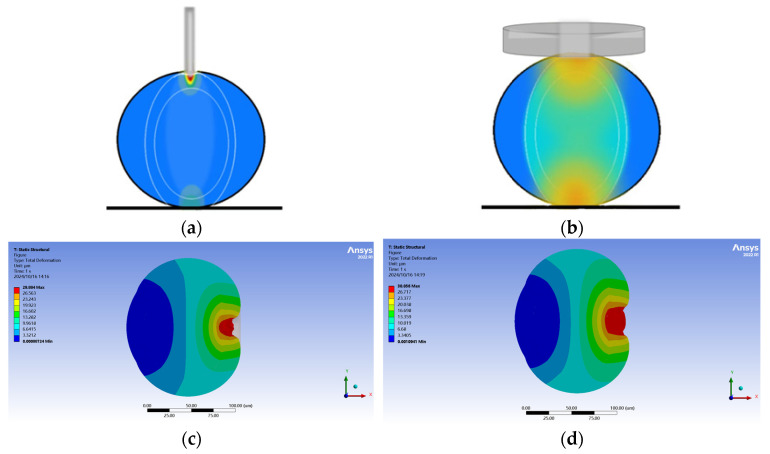
(**a**,**b**) Schematic diagrams of cell deformation cross–section when micropipettes with different sizes penetrate the cells. (**c**,**d**) Simulated cross–sections when micropipettes with different sizes penetrate the cells at 30 µm.

**Figure 8 sensors-24-06988-f008:**
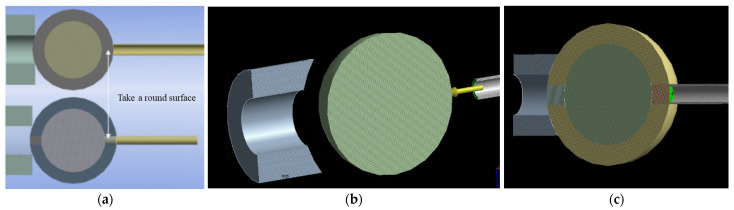
Comparison of models with different load components. (**a**) Structural diagram of different loads (**b**) Load acting on the micropipette (**c**) Load acting on the cell. (The green side of the figure is the position of the component applied load).

**Figure 9 sensors-24-06988-f009:**
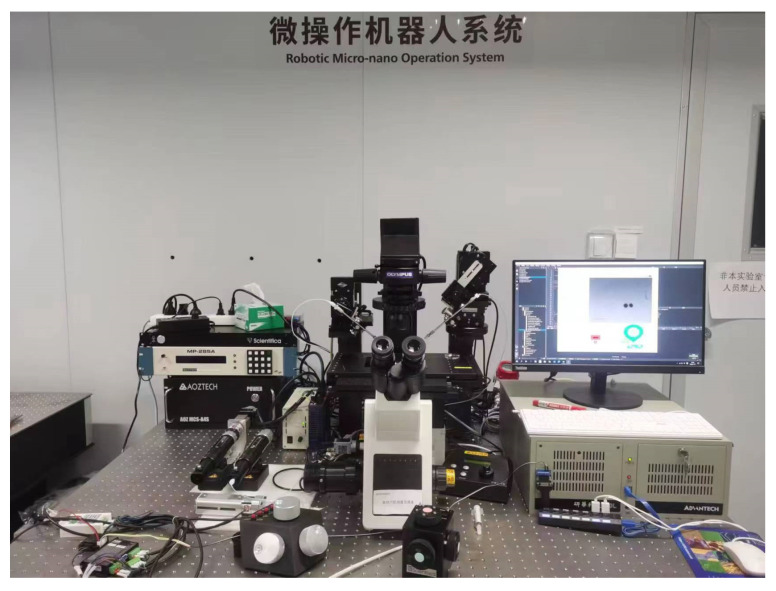
Experimental platform for cell penetration.

**Figure 10 sensors-24-06988-f010:**
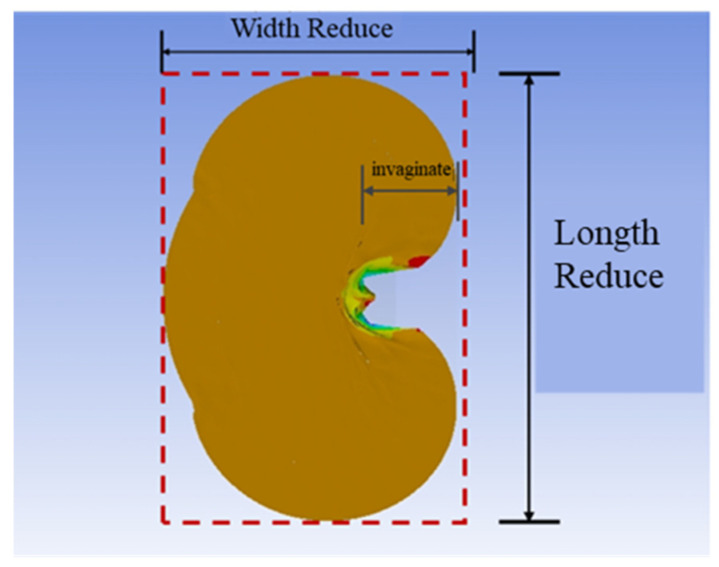
Schematic diagram of cell deformation parameters.

**Figure 11 sensors-24-06988-f011:**
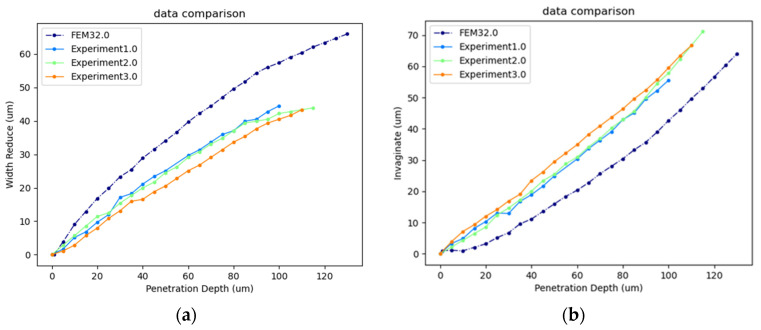
(**a**) Cell width reduction comparison between simulation and experiment. (**b**) Cell invaginate value comparison between simulation and experiment.

**Figure 12 sensors-24-06988-f012:**
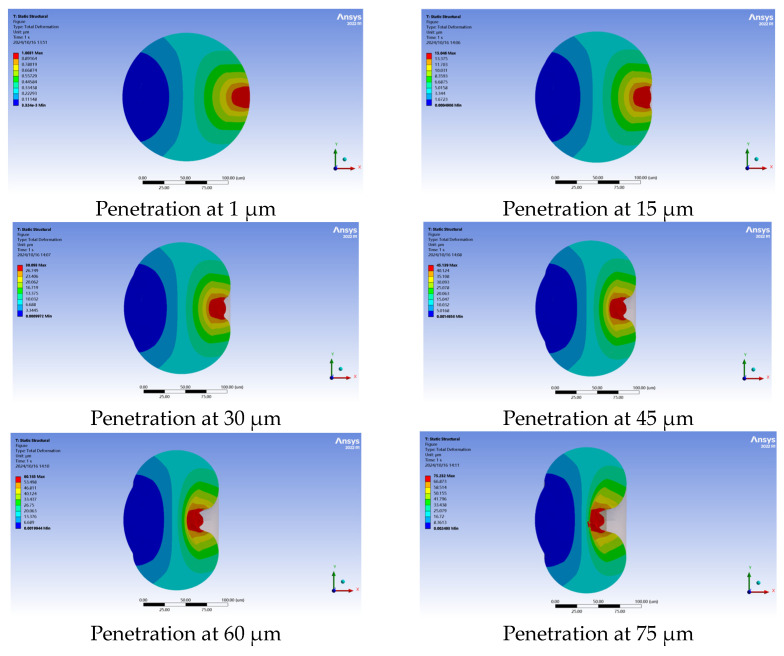
Cell deformation during cell penetration.

**Figure 13 sensors-24-06988-f013:**
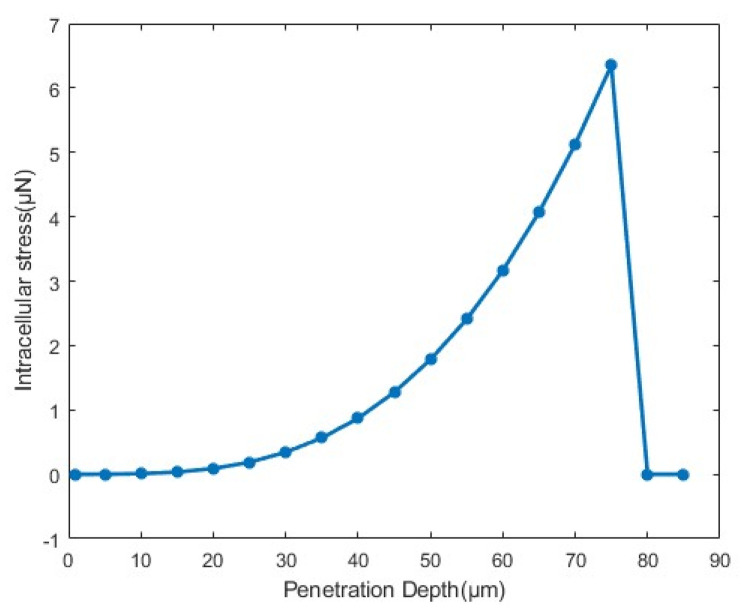
Stress–penetration depth curve obtained from simulation with the model setup in Section 3.4.1.

**Figure 14 sensors-24-06988-f014:**
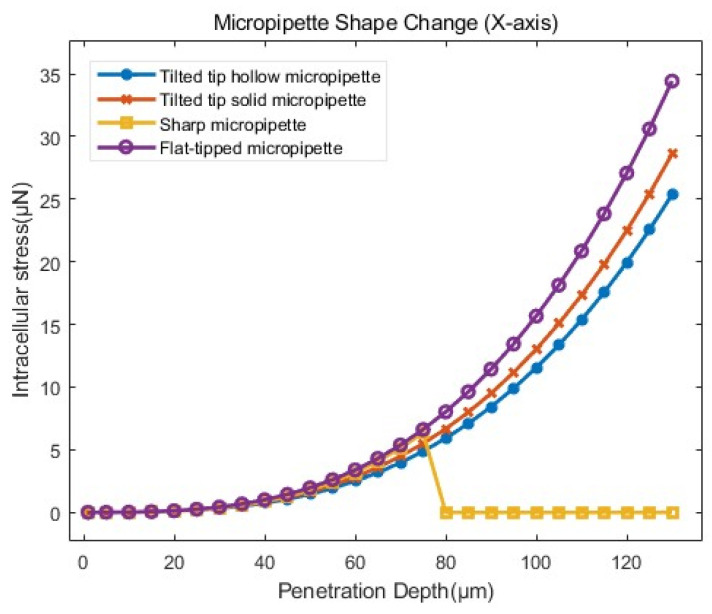
Stress–penetration depth curves from simulations with different shapes of micropipettes according to the model setup in Section 3.4.1.

**Figure 15 sensors-24-06988-f015:**
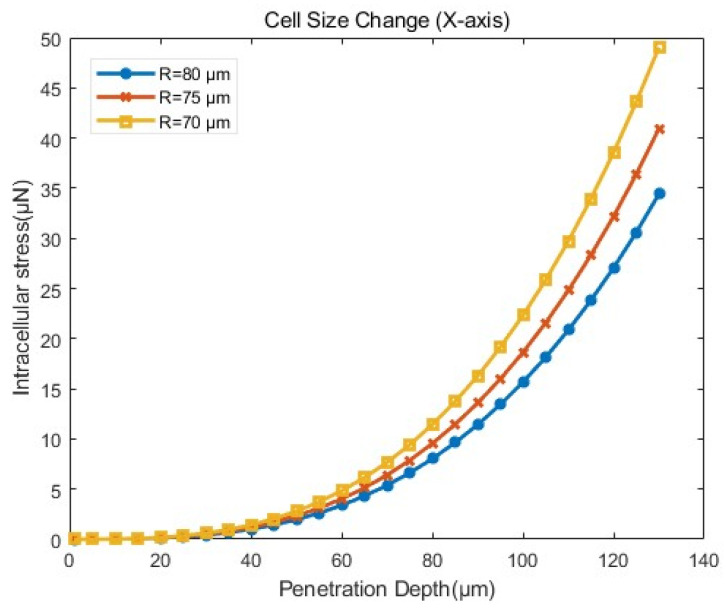
Stress–penetration depth curves from simulations with different sizes of cells according to the model setup in Section 3.4.1.

**Figure 16 sensors-24-06988-f016:**
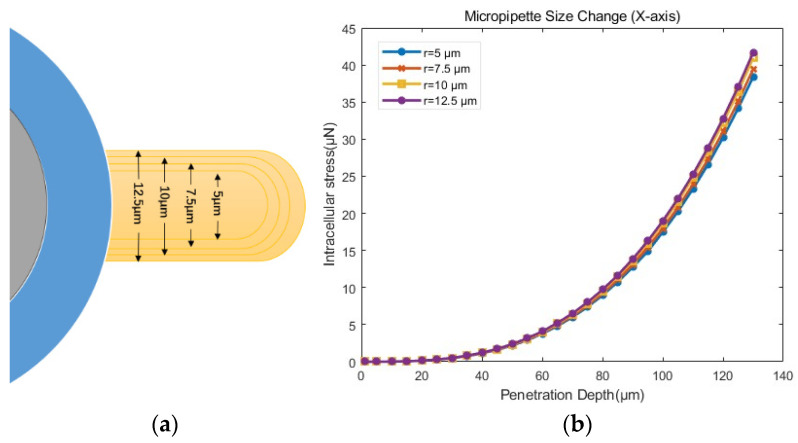
(**a**) Schematic diagram of micropipette size change (**b**) Stress–penetration depth curves from simulations with different radii of micropipette according to the model setup in Section 3.4.1.

**Figure 17 sensors-24-06988-f017:**
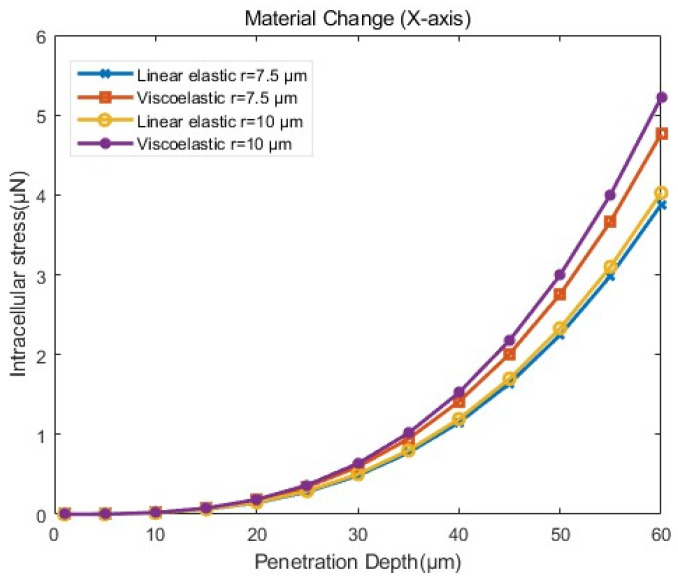
Stress–penetration depth curves from simulations with different materials of cells according to the model setup in Section 3.4.1.

**Figure 18 sensors-24-06988-f018:**
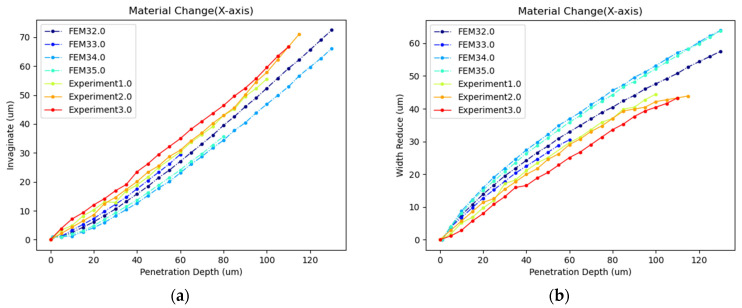
(**a**) Cellular invagination comparison between simulation and experiment with different cellular materials (**b**) Cellular width reduction comparison between simulation and experiment with different cellular materials (FEM32: linear elastic and r = 7.5 µm, FEM33: viscoelastic and r = 7.5 µm, FEM34: linear elastic and r = 10 µm, FEM35: viscoelastic and r = 10 µm).

**Figure 19 sensors-24-06988-f019:**
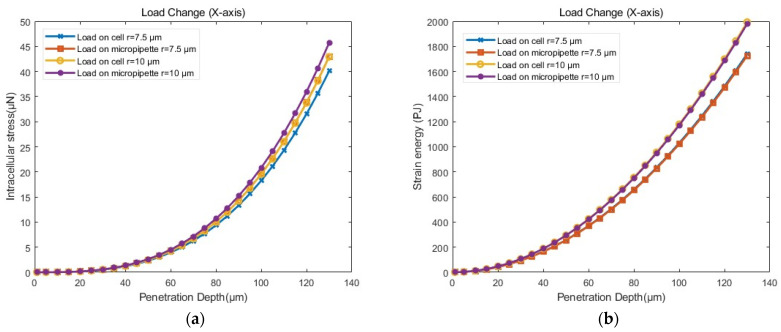
(**a**) Stress–penetration depth curves from simulations with different load components according to the model setup in Section 3.4.1. (**b**) Energy–penetration depth curves from simulations with different load components according to the model setup in Section 3.4.1.

**Figure 20 sensors-24-06988-f020:**
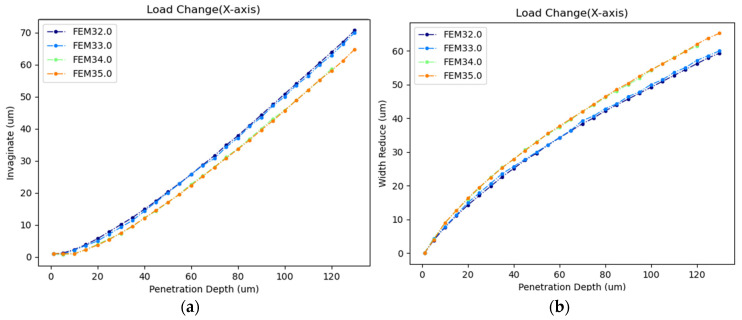
(**a**) Cellular invagination comparison with different load components (**b**) Cellular width reduction comparison with different load components (FEM32: load on cell and r = 7.5 µm, FEM33: load on micropipette and r = 7.5 µm, FEM34: load on cell and r = 10 µm, FEM35: load on micropipette and r = 10 µm).

**Figure 21 sensors-24-06988-f021:**
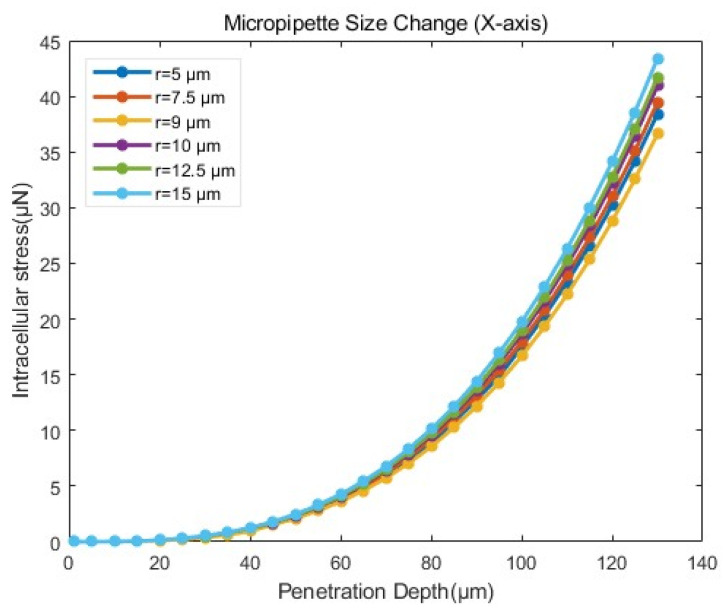
Stress–penetration depth curves from simulations for micropipettes of different radii according to the model setup in Section 3.4.1.

**Figure 22 sensors-24-06988-f022:**
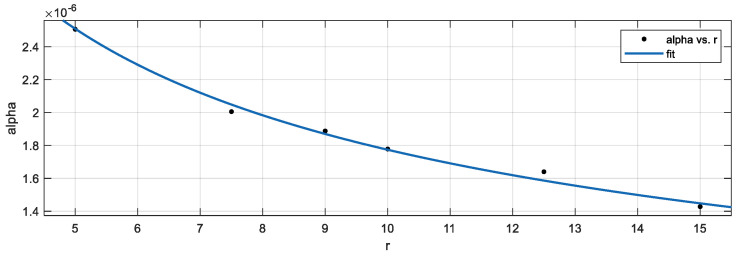
Relationship between micropipette radius r and α with the cell radius of 75 μm.

**Table 1 sensors-24-06988-t001:** Cell deformation curve fitting results.

Group	Width Reduction	Invaginate Value
Experiment 1.0	0.9968	0.9953
Experiment 2.0	0.9984	0.9970
Experiment 3.0	0.9959	0.9919

**Table 2 sensors-24-06988-t002:** The mean and standard deviation of the intracellular stress in penetration simulation of the influence of injection micropipette shape. The stress data before cells are penetrated (for a penetration depth d ranging from 1 to 75 μm) were used for statistical analysis.

Group		Average ± Standard Deviation (μN)
d (1–75 μm)	Tilted tip hollow micropipette	1.298 ± 1.475
	Tilted tip solid micropipette	1.464 ± 1.567
	Sharp micropipette	1.642 ± 1.387
	Flat–tipped micropipette	1.763 ± 1.719

**Table 3 sensors-24-06988-t003:** The mean and standard deviation of the intracellular stress in penetration simulation of the influence of cell size.

Group	Average ± $\mathrm{Standard}\;\mathrm{Deviation}\;\left(\mu N\right)$
R=70 μm	12.73 ± 3.681
R=75 μm	10.63 ± 3.367
R=80 μm	8.936 ± 3.208

**Table 4 sensors-24-06988-t004:** The mean and standard deviation of the intracellular stress in penetration simulation of the influence of micropipette size.

Group	Average ± $\mathrm{Standard}\;\mathrm{Deviation}\;\left(\mu N\right)$
r=5 μm	9.969 ± 3.389
r=7.5 μm	10.24 ± 3.434
r=10 μm	10.63 ± 3.499
r=12.5 μm	10.82 ± 3.531

**Table 5 sensors-24-06988-t005:** The mean and standard deviation of the intracellular stress in penetration simulation for the influence of cellular viscoelastic material were calculated. The stress data before cells penetration (for a penetration depth d ranging from 1 to 60 μm ) were used for statistical analysis.

Group		Average ± $\mathrm{Standard}\;\mathrm{Deviation}\;\left(\mu N\right)$
d (1–60 μm)	linear elastic–r=7.5 μm	1.050 ± 1.108
	viscoelastic–r=7.5 μm	1.289 ± 1.228
	linear elastic–r=10 μm	1.090 ± 1.129
	viscoelastic–r=10 μm	1.402 ± 1.284

**Table 6 sensors-24-06988-t006:** The mean and standard deviation of the intracellular stress and strain energy in penetration simulation of the influence of load component.

Group		Average ± Standard Deviation
Intracellular Stress $\left(\mu N\right)$	load on micropipette–r=7.5 μm	10.41 ± 3.463
	load on cell–r=7.5 μm	11.16 ± 3.585
	load on micropipette–r=10 μm	11.14 ± 3.582
	load on cell–r=10 μm	11.87 ± 3.698
Strain Energy $\left(PJ\right)$	load on micropipette–r=7.5 μm	591.4 ± 23.23
	load on cell–r=7.5 μm	586.3 ± 23.13
	load on micropipette–r=10 μm	677.4 ± 24.86
	load on cell–r=10 μm	672.5 ± 24.77

**Table 7 sensors-24-06988-t007:** Fitting results of parameter  α.

Radius of Micropipette r (μm)	α	$R^{2}$
5	$12.55\times 10^{-6}$	0.9993
7.5	$15.07\times 10^{-6}$	0.9995
9	$17.01\times 10^{-6}$	0.9999
10	$17.80\times 10^{-6}$	0.9999
12.5	$20.50\times 10^{-6}$	0.9996
15	$21.45\times 10^{-6}$	0.9998

**Table 8 sensors-24-06988-t008:** Fitting results of parameters α1 and β.

Group	Radius of Cell R (μm)	α1	β	$R^{2}$
1	70	$8.270\times 10^{-6}$	0.4227	0.9919
2	75	$5.613\times 10^{-6}$	0.4997	0.9981
3	80	$6.520\times 10^{-6}$	0.3635	0.9969

## Data Availability

The raw data supporting the conclusions of this article will be made available by the authors without undue reservation.

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
