# Peer review of "Modeling and Evaluation of Penetration Process Based on 3D Mechanical Simulation"

_sensors, 2024, doi:10.3390/s24216988_

Round 1
Reviewer 1 Report
Comments and Suggestions for Authors
The authors presented a cell penetration model based on the three-dimensional mechanical simulation and evaluated the internal stress during penetration , which is interesting. However, there are some issues to be clearly addressed before publication.
1 the latest researches on cell penetration are lack of discussion in the introduction section.
2 What are the advantages of the reported cell penetration model compared with existing models.
3 The image layout is quite chaotic, and some images lack numbering, such as Figure 13.
4 Which mathematical method is used in the simulations? Please describe the equations in the paper.
5 English has to be very carefully corrected.
Author Response
We gratefully thank the editor and all reviewers for their time spend making their constructive remarksand useful suggestions, which has significantly raised the quality of the manuscript and has enable us to improve the manuscript. Each suggested revision and comment brought forward by the reviewers was accurately incorporated and considered. Below the comments of the reviewers are response point by pointand the revisions are indicated.
Please see the attachment.

Reviewer 2 Report
Comments and Suggestions for Authors
The core focus of this paper is the development and validation of a finite element model (FEM) to simulate and optimize the cell penetration process, to reduce cell deformation and internal stress, thereby minimizing cell damage. The study investigates the impact of parameters such as the shape and radius of injection micropipettes, cell size, and material properties on the penetration process. By comparing simulation results with experimental data, the paper proposes strategies to optimize cell penetration, enhancing the efficiency and safety of micromanipulation procedures. Specifically, the model is built using ANSYS software, and both stress and deformation in cells under various conditions are analyzed and experimentally validated.
However, some minor points need to be paid attention to.
1. The experimental validation is based on several samples (e.g., three oocytes). Although the fit between the simulation and experimental data is high, the limited sample size may affect the model's generalizability.
2. Although the paper compares the effects of different load positions, it does not fully explore the variations in stress distribution caused by different micropipette contact methods. The position of the load could have additional effects on cell damage, especially in more complex operational environments.
3. The paper employs various complex sentence structures, demonstrating the author's solid command of English grammar. Although the writing is of a high standard, there is room for improvement in terms of conciseness and avoiding verbosity. For instance, some descriptions could be more succinct, particularly when presenting the experimental methods and results. Streamlining sentences and avoiding repetitive phrasing would further enhance the readability and professionalism of the paper.
Author Response
We feel great thanks for your professional review work on our article. As you are concerned, there are several problems that need to be addressed. According to your nice suggestions, we have made extensive corrections to our previous draft, the detailed corrections are listed below.
Please see the attachment.

Reviewer 3 Report
Comments and Suggestions for Authors
Cell penetration is a critical step in cloning and has an important impact on the developmental potential of cells. Minimizing cell damage during micromanipulation, particularly during injection and other manipulations, presents a substantial challenge. Therefore, the authors used finite element simulation to comprehensively analyze the puncture process by taking into account a variety of influencing factors, such as microtubule shape, cell size, and material properties. An internal stress model was developed to guide the selection of suitable micropipettes. The paper is well-structured, featuring a wealth of graphs and figures, and the data are thoroughly analyzed, providing a robust foundation for understanding the mechanics of cell penetration. However, there are some problems, which must be solved before it is considered for publication.
1. The format of numbers and units is not harmonized. Like line 290 "Oocytes are extracted from follicles with a diameter of 2 to 6 mm.", and lines 329 and 330 "Young's modulus of the cytoplasm is 784 Pa, whose radius is 55μm. Young's modulus of the ZP was 20 kPa, whose outer radius is 75μm, and the thickness is 20μm.". Problems like these are found throughout the manuscript and revisions are strongly recommended.
2. The figure notes in Figure 5, Figure 7 and Figure 12 are too blurry. Please consider replacing them with clearer ones.
3. The format of the figure notes is not harmonized. For example, the figure note in the upper left corner of Figure 17 is not boxed, while the figure note in Figure 18 is boxed, so please harmonize the format.
4. More statistical descriptions of the distribution of the data, such as sample size (n), mean, standard deviation are highly recommended.
5. The authors could explore data on the long-term effects of cells after penetration, such as the ability of cells to divide and differentiate.
Comments on the Quality of English LanguageThe grammatical presentation, terminology and structure of this article are fluent. It only requires a few minor adjustments in details to further enhance its quality and readability. Long sentences should be streamlined, coherence between paragraphs should be strengthened, and the innovative aspects of the study should be highlighted.
Author Response
Thank you for your nice comments on our article. According to your suggestions, we have supplemented several data here and corrected several mistakes in our previous draft. Based on your comments, we also attached a point-by-point letter to you. The detailed point-by-point responses are listed below.
Please see the attachment.
